# Biallelic Loss of Function Mutation in Sodium Channel Gene *SCN10A* in an Autism Spectrum Disorder Trio from Pakistan

**DOI:** 10.3390/genes13091633

**Published:** 2022-09-11

**Authors:** Ansa Rabia, Ricardo Harripaul, Anna Mikhailov, Saqib Mahmood, Shazia Maqbool, John B. Vincent, Muhammad Ayub

**Affiliations:** 1Molecular Neuropsychiatry & Development (MiND) Lab, Campbell Family Mental Health Research Institute, Centre for Addiction and Mental Health, Toronto, ON M5T 1R8, Canada; 2Department of Human Genetics & Molecular Biology, University of Health Sciences, Lahore 54600, Pakistan; 3Department of Anatomy, Institute of Dentistry, CMH Lahore Medical College, Affiliated with National University of Medical Sciences, Rawalpindi 46000, Pakistan; 4Institute of Medical Science, University of Toronto, Toronto, ON M5S 1A8, Canada; 5Department of Developmental-Behavioral Pediatrics, University of Child Health Sciences & The Children’s Hospital, Lahore 54000, Pakistan; 6Department of Psychiatry, University of Toronto, Toronto, ON M5T 1R8, Canada; 7Department of Psychiatry, Queen’s University Kingston, 191 Portsmouth Avenue, Kingston, ON K7M 8A6, Canada; 8Department of Academic Psychiatry, University College London, London WC1E 6BT, UK

**Keywords:** autism spectrum disorder, consanguinity, Pakistan, recessive inheritance, neurodevelopmental disorders

## Abstract

The genetic dissection of autism spectrum disorders (ASD) has uncovered the contribution of de novo mutations in many single genes as well as de novo copy number variants. More recent work also suggests a strong contribution from recessively inherited variants, particularly in populations in which consanguineous marriages are common. What is also becoming more apparent is the degree of pleiotropy, whereby mutations in the same gene may have quite different phenotypic and clinical consequences. We performed whole exome sequencing in a group of 115 trios from countries with a high level of consanguineous marriages. In this paper we report genetic and clinical findings on a proband with ASD, who inherited a biallelic truncating pathogenic/likely pathogenic variant in the gene encoding voltage-gated sodium channel X alpha subunit, *SCN10A* (NM_006514.2:c.937G>T:(p.Gly313*)). The biallelic pathogenic/likely pathogenic variant in this study have different clinical features than heterozygous mutations in the same gene. The study of consanguineous families for autism spectrum disorder is highly valuable.

## 1. Introduction

Heterozygous missense mutation in the neuronal voltage-gated sodium channel X alpha subunit (also known as voltage-gated sodium channel subunit alpha Nav1.8) gene, *SCN10A*, were reported to be the cause of familial episodic pain syndrome-2 (FEPS2; [1]). More recently, common and rare heterozygous missense variants in *SCN10a* have been reported for Brugada syndrome (BrS), a cardiac arrhythmia syndrome with elevated risk of sudden cardiac death [2,3,4]. Heterozygous missense mutations in several other voltage-gated sodium channel genes have been associated with BrS (*SCN5A*: MIM 600163), familial episodic pain syndrome FEPS3 (*SCN11A*; MIM 604385), primary erythermalgia, and paroxysmal extreme pain disorder (*SCN9A*; MIM 603415). Heterozygous mutations, mainly missense, in other voltage-gated sodium channel genes are believed to cause developmental epileptic encephalopathies (*SCN1A*: MIM 182389; *SCN2A*: MIM 182390; *SCN3A*: MIM 182391; SCN8A: MIM 600702). In several reported instances, de novo loss of function mutations in *SCN2A* have been reported for ASD [5,6,7]. Interestingly, biallelic mutations in some of these genes cause quite distinct disorders. For instance, bialleic LoF mutations in *SCN9A* cause congenital pain insensitivity (MIM 243000).

We report here on a biallelic pathogenic/likely pathogenic variant in exon 7 of *SCN10A* (Chr3: 38802185C>A; NM_006514.2:c.937G>T; (p.Gly313*)), in an ASD proband, inherited from apparently healthy parents.

## 2. Methods

### 2.1. Ascertainment

The child was assessed at 52 months of age and recruited to the study from the Children’s Hospital and Institute of Child Health, Lahore, Pakistan.

### 2.2. DNA Extraction, Microarray and Sequencing

The DNA was extracted from venous blood using standard salting-out methods. In order to check for potentially pathogenic copy number variants (CNV), microarray analysis was performed on an Illumina iSCAN array scanner, using an Illumina CoreExome-24 v1.3 BeadChip (Illumina Inc., San Diego, CA, USA). The CNV calling was performed using Illumina Genome Suite/CNVpartition. Whole Exome Sequencing (WES) was performed on the DNA of the proband as well as both parents, using the Thruplex DNA-Seq (Rubicon Genomics, Ann Arbor, MI, USA) Library Preparation Kit with the Agilent SureSelect V5 Exome Capture kit, as described in Harripaul et al., submitted [5]. Sequencing was performed by The Centre for Applied Genomics (www.tcag.ca). Annotation and variant prioritization was performed as described in Harripaul et al., submitted [5], searching for potentially damaging homozygous variants, de novo heterozygous variants, and X-linked hemizygous variants. Variants of interest were confirmed initially through examining the WES reads using Integrated Genomic Viewer (IGV; https://software.broadinstitute.org/software/igv/) [8], and then through Sanger sequencing in parents and proband (www.tcag.ca).

## 3. Results

### 3.1. Clinical Assessment

The proband (IV-4) is a male child. His parents are double second cousins (see pedigree, Figure 1). He is fourth (out of four) in birth order and was born at full term after a normal vaginal delivery, with an APGAR score of 8 and birth weight of 2.2 kg. His mother took oral folic acid irregularly during pregnancy and she was anxious about the possibility of a cesarean delivery that she ultimately did not need.

There was no history of any psychiatric or neurodevelopmental disorder in the family. The three elder siblings were all reported as unaffected but were unavailable for genetic studies. Both the parents were healthy and there was no history of sudden cardiac death in the family. The history, neurological and cardiac examination and ECG did not reveal any neurological or cardiac abnormality among the parents. At 24 months of age his parents sought help because of concerns about his development. He did not respond to his name, had limited eye contact and only 4–5 words of speech at that stage.

At 30 months of age on administration of the Modified Checklist for Autism in Toddlers (M-CHAT) questionnaire [9] to the accompanying parents, he was categorized as ‘at risk’ for autism. Additionally, on otoacoustic testing (performed at Department of Paediatric Audiology, The Children’s Hospital and The Institute of Child Health, Lahore, Pakistan) his hearing was normal.

At the age of 3 years (36 months), he was diagnosed with mild to moderate autism after assessment with Childhood Autism Rating Scale (CARS) [10], where he scored 36.5.

At 56 months of age an electrocardiogram (ECG) was normal.

An MRI was performed when he was 77 months old at Butt Hospital MRI Centre, Gujrat and a cavum velum interpositum was noted that was determined as a normal variant. An electromyogram and nerve conduction studies were performed on the proband at age 117 months years at the Combined Military Hospital Kharian and Kharian Medical College, Gujrat, Punjab. Nerve conduction studies using surface electrodes showed normal latency, sensory nerve action potential (SNAP) amplitudes and velocities. An electromyogram using concentric needle electrodes showed normal motor unit action potentials (MUAPs) and recruitment patterns, and showed no evidence of peripheral neuropathy or myopathy. There were no reported symptoms of insensitivity or hypersensitivity to pain.

The MA assessed him at age 118 months through an interview with parents and observations of his behavior. Based on his developmental history and current functioning, he is estimated to have severe intellectual disability. His developmental milestones were delayed. He does not utter any meaningful words. He is not toilet trained at this age. He is dependent on his parents for self-care. It is not feasible to administer an IQ test because of his level of disability. Based on the Diagnostic and Statistical Manual of Mental Disorders of American Psychiatric Association (DSM5) criteria he has severe intellectual disability [11].

He has been vaccinated as per government protocol and has not received MMR vaccine. The child is left-handed and is allergic to wheat items. He had complaint of insomnia initially and was hyperactive, however that had settled down after treatment with Risperidone (2 mg per day). There are no other co-morbidities in the child.

### 3.2. Dysmorphology Examination (Done at the Age of 4.4 Years)

Biometrics: normal occipitofrontal/head circumference; normal body weight = 15 kg (11–25th centile); short stature; normal body-mass index (BMI) = 16.6 (81st centile). Face/head: widow’s peak; prominent left ear; proximal root of nose with prominent columella; frontal bossing; long palpebral fissures; deep-set eyes; synophrys. Digits: left 5th finger clinodactyly.

In summary, apart from short stature, only subtle physical anomalies were observed.

### 3.3. Genomic Analysis

In a study of whole exome sequencing for a total of 115 ASD proband/mother/father trios from Pakistan, Iran and Saudi Arabia, we identified a biallelic pathogenic/likely pathogenic variant in exon 7 (out of 27) of the gene *SCN10A* (Chr3:38802185C>A; NM_006514.2:c.937G>T; p.(Gly313*); hg19 coordinates) [5] in this proband. The variant results in a substitution of a glycine codon for a stop codon at position 313. The variant when compared with the other rare variants was predicted to be deleterious (Appendix A). This resulting truncated protein would be lacking nineteen of the 24 predicted transmembrane domains, and with it would be three of four of the channel’s domains (Figure 2). Moreover, the variant is predicted to trigger nonsense-mediated mRNA decay (www.muatationtaster.org, accessed on 6 June 2021), and thus *SCN10A* mRNA levels would likely be reduced in addition to the functional loss for any resulting translated protein products. No other likely pathogenic CNVs, de novo variants or X-linked variants were identified for the proband.

## 4. Discussion

In this proband we identified a biallelic truncating variant in the gene *SCN10A*, encoding the voltage-gated sodium channel subunit alpha Nav1.8, that is a likely cause of his ASD and intellectual disability. The homozygous variant in *SCN10A* appears to be causing a phenotype that differs significantly from phenotypes reported for heterozygous mutations in the gene. This child did not have ECG or cardiac abnormalities, nerve conduction abnormalities or brain anatomical anomalies (MRI), and there were no problems with pain threshold. This gene has been reported to be associated with ASD before. In a Chinese patient a compound heterozygous mutation was identified. The patient had intellectual disability, ASD and pain insensitivity. The authors observed ASD like behaviors in a loss of function mouse model [12]. In another study an inherited heterozygous mutation was found in three siblings in SCN10A gene. In this family there were other mutations as well that were considered stronger candidates [13]. There are many other examples in the literature where the phenotype identified in heterozygous mutations is different from the one identified in homozygous mutations. For example, SCN9A, for which dominant mutations may cause paroxysmal extreme pain disorder (PEXPD; MIM 167400) [14], whereas biallelic mutations cause congenital insensitivity to pain (CIP; MIM 243000), which appears to be diametrically opposite phenotypes [15].

In the gnomAD database the prevalence of this variant is 0.000003979 and there are no reported homozygotes. The only variant allele at this position reported out of 251,296 alleles was also in a South Asian individual. GnomAD lists 306 Lof or probable LoF variants in the gene (in ~250,000 alleles), however only as heterozygotes, suggesting that it is tolerant to haplo-insufficiency but not to null (recessive LoF) variants (probability of being loss-of-function intolerant (*pLI*) score = 0). With so many heterozygous LoF variants in the population, one would anticipate an elevated chance of compound heterozygous null variants. However, examination of ASD whole genome sequence data (MSSNG: research.mss.ng), revealed no putative compound heterozygous null mutations were present in ~10,000 cases (accessed on 10 April 2022).

This finding further highlights the benefits of studying consanguineous families for neurodevelopmental disorders where the probability of identifying homozygous mutations is elevated.

## 5. Conclusions

This paper describes a recessive pathogenic/likely pathogenic variant in the SCN10A gene in a consanguineous family that results in a substantially different phenotype including ASD that has been reported previously for heterozygous mutations in this gene.

## Figures and Tables

**Figure 1 genes-13-01633-f001:**
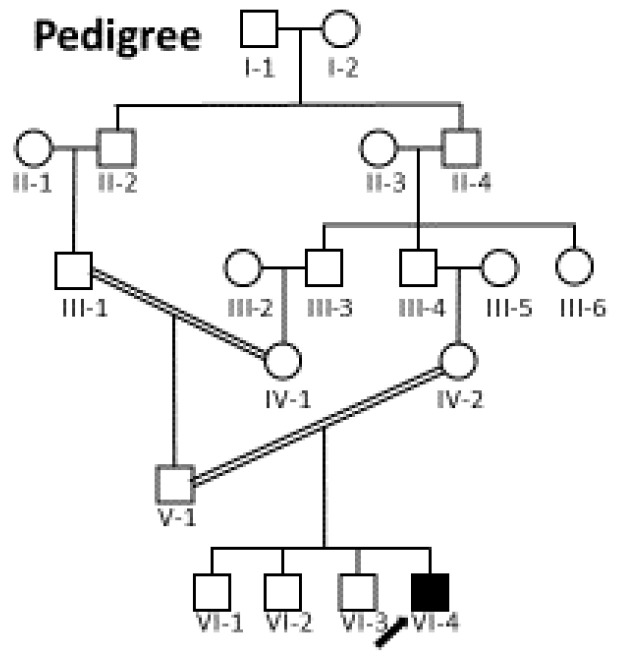
Pedigree. The proband is individual VI-4 and comes from a consanguineous marriage (parents are double second cousins). Black means affected and arrow indicates the proband recruited.

**Figure 2 genes-13-01633-f002:**
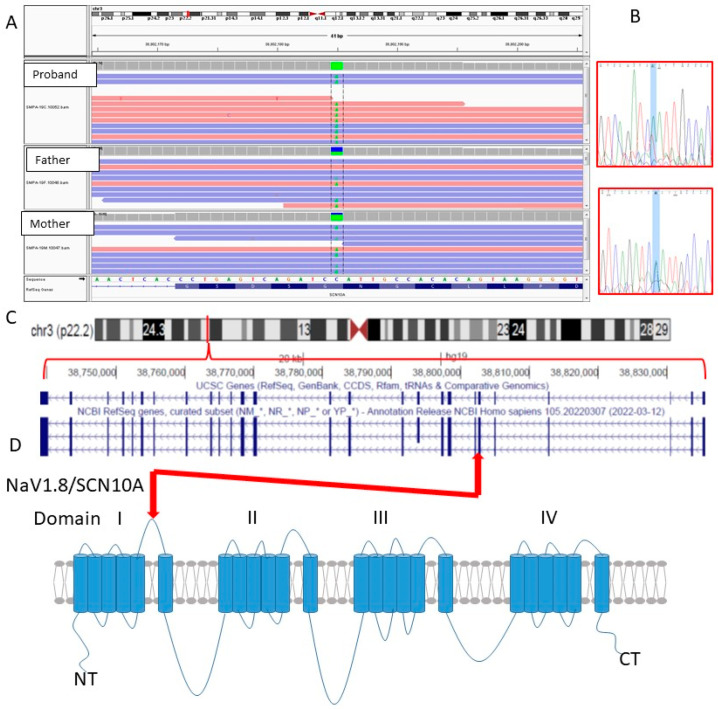
Validation and location of *SCN10A* variant: (**A**) Integrated Genomics Viewer (IGV, Broad Institute) view of whole exome sequencing reads for SMPA19 proband, mother and father, indicating location within *SCN10A* transcript. (**B**) Sanger sequence validation for proband and mother (insufficient DNA remaining from father), using FinchTV. Blue shading indicates location of Chr3:38802185C>A variant. (**C**) Ideogram of chromosome 3 indicating location of *SCN10A*, and genomic structure for the gene. (**D**) Representation of the SCN10A protein in a cell membrane, indicating transmembrane domains, and four main structural domains, I-IV, and position of the biallelic stopgain (nonsense) mutation p.(Gly313*) within the protein.

## Data Availability

The datasets generated and analyzed during the current study are not publicly available because the sample size is small and there is a risk that the individual identity can be inferred from the genomic data but are available from the corresponding author on reasonable request.

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
