# Peer review of "Biallelic Loss of Function Mutation in Sodium Channel Gene SCN10A in an Autism Spectrum Disorder Trio from Pakistan"

_genes, 2022, doi:10.3390/genes13091633_

Round 1

Reviewer 1 Report

In this paper, Rabia et al. describe a biallelic loss of function mutation in sodium channel gene SCN10A in an autism spectrum disorder trio from Pakistan. Although this is an interesting report, providing new information for the pathogenesis of ASD, considering its content and scope, I would recommend it as a communication rather than an article.

There following concerns should be addressed:

The Abstract should focus more on the content of the paper and it will helpful if some comparisons are made later in Discussion with other ASD-causing genes.

In the Results at the Clinical assessment additional information is required on family history, particularly heart disease, or sudden (cardiac)/unknown death.  In the patient’s history to follow the chronological order is very important with entering the child's age in month/year at the time of the examination, or disease event, the dates have to be deleted. As that the mutation is VUS/LP, it is therefore very important to perform a cardiological and neurological examination of the parents in order to detect possible mild symptoms. These data must also be described in this section. 

At the Dysmorphologic examination, the standard deviations (SDs) at the anthropometric data and standardized terms to describe the morphologic symptoms (e.g. use synophris instead of eyebrows: synophris) should be used. This will make the paper much readable.

At the Genomic analysis, “whole exome sequencing” is the correct term instead of the used “whole exome sequence”. It is not clear, that the proband mentioned in this section is the same patient mentioned in the previous sections? It would be interesting and important to know if there were other patients with SCN10A mutations in the WES study cohort of 115 ASD patients. To avoid confusion in the description of genetic variant authors should follow the conventional HVGS nomenclature recommendations throughout in the manuscript. Please, indicate the clinical significance of the mutation based on the conventional databases. Figure 2 will be the appropriate mention later in this section.

I suggest writing a short comparison with other ASD-causing genes in the Discussion. It will significantly improve the value of article, as it is mentioned above. It will be important to explain in a more detail manner, why the Authors consider the found variant to be pathogenic.

Adding further recent references to the paper in order to do it more-comprehensive is a good strategy. Hereby is a suggestion of two important articles from this field:

-  Heinrichs B, Liu B, Zhang J, Meents JE, Le K, Erickson A, Hautvast P, Zhu X, Li N, Liu Y, Spehr M, Habel U, Rothermel M, Namer B, Zhang X, Lampert A, Duan G. The Potential Effect of Na v 1.8 in Autism Spectrum Disorder: Evidence From a Congenital Case With Compound Heterozygous SCN10A Mutations. Front Mol Neurosci. 2021 Jul 27;14:709228. doi: 10.3389/fnmol.2021.709228. PMID: 34385907; PMCID: PMC8354588.

- Dhaliwal J, Qiao Y, Calli K, Martell S, Race S, Chijiwa C, Glodjo A, Jones S, Rajcan-Separovic E, Scherer SW, Lewis S. Contribution of Multiple Inherited Variants to Autism Spectrum Disorder (ASD) in a Family with 3 Affected Siblings. Genes (Basel). 2021 Jul 8;12(7):1053. doi: 10.3390/genes12071053. PMID: 34356069; PMCID: PMC8303619.

Because of the numerous grammatical and stylistic mistakes the scientific value of the paper seems to loose much and it makes difficulties in following the main topic. Please correct grammatically/typographically the errors throughout the manuscript.

Author Response

Please see the attached point by poit response. 

Reviewer 2 Report

This is an interesting case report, especially given the high level of consanguinity and the rarity of the variant.

Major comments:

My major concern is if the right variant have been identified as causative. Especially consanguineous families have a high level of homozygous variants. Additionally, the siblings were not available for genetic testing. Related to this issue I have a few questions:

1.Were any other family members available for genetic testing?

2. What were the variant filtering criteria?

3. Were any other homozygous or heterozygous variants associated with ASD identified? Please present a supplement a list of the rare variants obtained after the filtering.

4. Was anybody else from the family affected with ASD or had any neuropsychiatric symptoms?

5. Is there any possibility to perform mRNA studies?

Other comments:

1. Introduction: Please mention that compound heterozygous SCN10A variants have been associated with ASD (https://doi.org/10.3389/fnmol.2021.709228)

2.  Was the study approved by an Ethic committee (if yes, please provide the consent number)?

3. Was sequencing performed in house or through an external company? If it was performed through an external company please provide its name.

4. Line 95: Please provide the age of the particular milestones (may be in a table)

5. Line 103-104: please provide Risperidone doses. Were any others medication tried?

6. Line 146 and 147 please provide such examples.

7. Please compare in the discussion the reported phenotypes with other ASD phenotypes reported for SCN10A (https://doi.org/10.3389/fnmol.2021.709228)

Author Response

Please see attached responses.

Round 2

Reviewer 2 Report

I am satisfied with the answers. I would just replace the word "mutation" with "pathogenic/likely pathogenic variant"..